# Fuzzy Cognitive Scenario Mapping for Causes of Cybersecurity in Telehealth Services

**DOI:** 10.3390/healthcare9111504

**Published:** 2021-11-05

**Authors:** Thiago Poleto, Victor Diogho Heuer de Carvalho, Ayara Letícia Bentes da Silva, Thárcylla Rebecca Negreiros Clemente, Maísa Mendonça Silva, Ana Paula Henriques de Gusmão, Ana Paula Cabral Seixas Costa, Thyago Celso Cavalcante Nepomuceno

**Affiliations:** 1Departamento de Administração, Universidade Federal do Pará, Belém 66075-110, Brazil; thiagopoleto@ufpa.br (T.P.); ayara.ufpa@gmail.com (A.L.B.d.S.); 2Campus do Sertão, Universidade Federal de Alagoas, Delmiro Gouveia 57480-000, Brazil; victor.carvalho@delmiro.ufal.br; 3Centro Acadêmico do Agreste, Universidade Federal de Pernambuco, Caruaru 55014-900, Brazil; tharcylla.clemente@ufpe.br; 4Departamento de Engenharia de Produção, Universidade Federal de Pernambuco, Recife 52171-900, Brazil; maisa@cdsid.org.br (M.M.S.); apcabral@cdsid.org.br (A.P.C.S.C.); 5Departamento de Engenharia de Produção, Universidade Federal de Sergipe, Aracaju 49100-000, Brazil; anapaulagusmao@cdsid.org.br

**Keywords:** cybersecurity, fuzzy cognitive maps, telehealth, scenario analysis, planning

## Abstract

Hospital organizations have adopted telehealth systems to expand their services to a portion of the Brazilian population with limited access to healthcare, mainly due to the geographical distance between their communities and hospitals. The importance and usage of those services have recently increased due to the COVID-19 state-level mobility interventions. These services work with sensitive and confidential data that contain medical records, medication prescriptions, and results of diagnostic processes. Understanding how cybersecurity impacts the development of telehealth strategies is crucial for creating secure systems for daily operations. In the application reported in this article, the Fuzzy Cognitive Maps (FCMs) translated the complexity of cybersecurity in telehealth services into intelligible and objective results in an expert-based cognitive map. The tool also allowed the construction of scenarios simulating the possible implications caused by common factors that affect telehealth systems. FCMs provide a better understanding of cybersecurity strategies using expert knowledge and scenario analysis, enabling the maturation of cybersecurity in telehealth services.

## 1. Introduction

The Brazilian Ministry of Health created the national telehealth system in 2007 with the initial objective of promoting family health remotely by using Information and Communication Technologies (ICT). One factor that justifies implementing this system is delivering healthcare to people living in remote communities where the nearest hospital care is distant. Bernardes et al. [1] stated that based on data from the Brazilian Institute of Geography and Statistics, only 24% of the country’s population live in large cities, which adds to telehealth’s importance as a public policy.

During the first semester of 2020, telehealth, also called telemedicine strategies, became essential in Brazil and many other countries due to the COVID-19 pandemic pressure on the limited hospital resources and the related response from public authorities imposing quarantine campaigns and mobility interventions worldwide. According to Nepomuceno et al. [2], when many potentially infected patients require regular or intensive care at the same time, hospitals with limited resources end up overloaded, the probability of propagation increases, and, as a result, the health systems collapse due to the lack of technical resources, fatigue, and overloading health teams. COVID-19 lockdown and social distance strategies in many have presented an opportunity for both doctors and patients to use telemedicine as a new manner of engagement and treatment in many regions [3,4].

The Telehealth Guidelines established by the Ministry of Health through Decree-Law No. 9795, of 17 May 2019, are mainly intended to improve user satisfaction and the quality of services provided to citizens through the Unified Health System [5]. The related systems have confidential data such as patient health histories, drug prescriptions, and medical diagnoses. Such data can be the target of cyberattacks, highlighting the importance of well-defined strategies for their protection. According to Kruse et al. [6], there was a 22% increase in cyberattacks in 2015, compromising about 112 million medical information records.

It is emphasized that cybersecurity should not be analyzed only as a compliance practice given the occurrence of specific events causing additional costs [7,8], but should be designed in a structured and contingent way to consider all systems from the conception of telemedicine systems and services to be offered [9,10]. Deficiencies in the ICT infrastructure of these services contribute significantly to the increase of harmful attacks on health organizations that also adopt the strategy of promoting their services remotely [11]. Thus, the ICT infrastructure is a crucial factor in developing cybersecurity analysis to implement telehealth systems [12,13,14,15]. The importance of considering vulnerabilities is often associated with the risk of losses, corruptions, inappropriate changes, and theft of data, with information and documents that affect the integrity of medical diagnoses delivered to the patient, which can cause serious damage to the health of the individual [16]. In general, these situations allow threats to be exploited and are often caused by cyberattacks from malicious systems or people [17]. Zain et al. [18] identified four main situations verified in cyberattacks which can occur in telehealth services, such as (i) when the data is destroyed or becomes unavailable, (ii) when an unauthorized system or person accesses the database, (iii) when an unauthorized system or person obtains access to the service and makes improper changes, and (iv) when an unauthorized system or person inserts counterfeit objects into the database. These situations are possible failures or threats in the data transmission process, which can be accidental or purposeful.

In telehealth services, the main challenge of the physicians is protecting the privacy of data. However, most of these professionals do not receive adequate training, and they are subject to situations that may compromise the performance of healthcare. This context requires preventive actions and security tools due to the sensitive data in healthcare systems such as digital signatures, professional credentials, financial data, patient diagnostic images, among others [19]. It is worth mentioning that this concern becomes even more complex when considering cyberattacks, especially due to the different interactions that occur on the Internet [20]. Furthermore, failure to comply with legal regulations may result in financial or criminal penalties [21,22]. For this, the IT professionals must make strategic decisions to define security policies and ensuring authenticity, integrity, and confidentiality of the database, besides ensuring business sustainability. 

Little research has been carried out in the context of cybersecurity in telehealth and on attacks on related systems to analyze the damaging effects of information stored on patients’ clinical health. Poleto et al. proposed a framework for cybersecurity risk management in telemedicine [23]. New studies focus can be oriented towards cybersecurity aspects, determining causal relationships either to prevent attacks or to solve problems that have already occurred, ensuring the security of services and, consequently, the activities and associated practices. The use of tools to support the identification of these security factors in telehealth services is beneficial for this purpose; however, the analytical process can be complex, and it requires high cognitive effort from the professionals involved, whether analysts or decision-makers, towards the planning of different assessment scenarios, helping to choose the best security measures.

Most of these strategic decisions are involved in business sustainability process [24], which can define action plans to ensure the telehealth services operation. The ICT management process assists in directing how medical centers can use IT to manage technologic solutions. For this, it is opportune to present methodologies to support organizational diagnoses to identify these possible causes of threats in telehealth systems. One of these methodologies is Fuzzy Cognitive Maps (FCMs) [25], which represents scientific knowledge and strategic decision making in systems using elements of a mental map, based on fuzzy logic computation.

This context into account, this article proposes an analytical approach based on Fuzzy Cognitive Maps (FCM) aimed at the mental representation of experts on causal relationships within a set of concepts related to cybersecurity that impact telehealth systems, providing support for strategic planning and decision-making. FCM can represent all relationships intelligibly, enabling creating scenarios and reducing cognitive effort by allowing their analysis through objective graphic elements, and representing interesting support to improve information asset protection concerning patient information management. This article aims to demonstrate the results of applying FCMs in favor of cybersecurity in a telehealth system, seeking to identify variables that can be used for cybersecurity planning, in addition to simulating involved scenarios. The remaining of this paper is organized as follows: Section 2 presents the Materials and Methods, explaining the mechanism of the proposed approach. Section 3 undertakes an application that validates the proposed approach. Section 4 is the discussion of the main findings; conceptual and practical implications are in Section 5. Finally, Section 6 draws some conclusions, indicates some study limitations, and suggests future research lines.

## 2. Materials and Methods

According to Tsadiras [26], FCM analysis allows identifying strategies cybersecurity in a system having a more significant impact on other factors and provides possible scenarios by varying the degree of intensity of these variables in a complex problem. Moreover, incorporating the subjectivity and knowledge of an expert leads to a constructivist methodology and provides a complement to information security planning in hospitals.

Protecting patients’ private data in telehealth services can be severely damaged by malicious interventions, such as altering or stealing data and information. Other factors, such as data privacy and credibility, can affect the image of the medical center. In Brazil, telehealth services have been valued in recent years and this has encouraged governmental decisions regarding (i) the prioritization of telemedicine infrastructure; (ii) the systematization of the teleassistance process, with the development of clinical data cybersecurity protocols; and (iii) the structuring of security planning to provide the quality and confidentiality of the data and services offered by telehealth in hospitals.

The present research’s motivation is based on the following question: what are the main cybersecurity factors affecting telehealth? In response to this question, the following issues will be discussed: (i) the role of stakeholders in the cybersecurity decision process at a hospital; (ii) the use of FCM as an integrated methodology to analyze cybersecurity, to develop planning policies, and to assess the impacts of such decisions in hospital.

First, we identified the main security concepts that occur in telehealth services. For this, an informative and analytical list of concepts that may influence cybersecurity planning in telehealth at a hospital was created. Considering that the planning decisions are strategic, a manager in the ICT area of a hospital assumed the expert’s role in eliciting the concepts in the cybersecurity context. Two technical meetings were held with the hospital’s ICT manager, each having an average duration of two hours, coordinated by a facilitator who is an expert in information security and responsible for analyzing the results. During the interview, the study’s objectives and the research procedure were presented, allowing for a better understanding of the study by the ICT manager. As a result, the list of the main concepts and the description of the leading information about security strategies adopted to treat and prevent problems caused by cyberattacks in telehealth services were obtained, considering the ICT manager’s perception [27]. 

This list consisted of grouping the concepts that affect cybersecurity and analyzing the cause and effect relationship between them. For this, the Mental Modeler software was used to obtain the expert’s cognitive map [28]. The ICT manager identified causal connections between the nodes, which required defining the type of relationship (positive or negative), between *w_i_* and *w_j_*, and the intensity of each one over the other. The dynamic analysis of the FCM focuses on evaluating the system’s behavior when the cause and effect relationships between the selected concepts are changed, enabling the evaluation of different scenarios [29].

The information was collected to support developing a strategic plan dedicated to cybersecurity in telehealth at hospital. Moreover, to analyze the changes that may impact cybersecurity, the construction of scenarios involves using the identified relationships among the concepts. Consequently, the scenarios can be considered roadmaps for developing and improving the model that describes the problem in a learning process. This study’s cognitive structure allowed for greater transparency in cybersecurity planning of telehealth services and theoretical contributions, directed to strategic decisions, and promoting organizational learning [30].

### FCM Procedure

A FCM can be described as a fuzzy graph containing the concepts to be casually assigned in the nodes and the relationships in the edge arrows [25]. The procedure for creating the FCM can be defined in three main steps [27]:

*First Step*: clarify the FCM purpose and if it is not well defined the search for causal relationships will make the formation of the FCM unfeasible.

*Second Step*: identify the relevant concepts that influence the decision to be taken.

*Third Step*: find the causal relationships between the concepts defined in the previous step, so that these relationships need to be abstracted from the decision maker’s definitions, through instruments such as questionnaires and interviews.

Thus, from a mathematical point of view, an FCM can be described as a set of nodes (concepts) *C_i_* with *i* = 1, …, *n*, being the number of concepts in the problem and all these concepts together represent a vector of state *A* = [*A*_1_, …, *A_n_*]. The value of each concept is influenced by the values of the concepts that are related to it along with the corresponding causal weight and for the concept system to evolve, the vector *A* needs to be passed repeatedly over the connection matrix *W* [31].

The associated mathematical formula is given in Equation (1) [32]:(1)$$A_{I}^{\left(K+1\right)}=f\left(A_{i}^{k}+\overset{N}{\underset{\begin{array}{c} j\neq i\\ j=1 \end{array}}{\sum }}A_{j}^{k}W_{ji}\right)$$
where:$A_{I}^{\left(K+1\right)}$ is the value of concept *C_i_* at step *k* + 1;$A_{j}^{k}$ is the value of the concept *C_j_* in step *k*;$W_{ji}$ is the weight of the relationship between *C_j_* and *C_i_*; andf(x) is a sigmoid threshold function defined by Equation (2):
(2)$$f=\frac{1}{1+e^{-\lambda x}}$$
where *λ* is a positive constant in a determined interval and *f*(*x*) lies between [0, 1].

## 3. Results

The proposed FCM model considers a holistic view to analyze cybersecurity concepts within telehealth in a hospital in the Amazon region. In the model, minimal changes were necessary to expand the notion and technical specifications for adequate cybersecurity planning. First, the concept of cybersecurity was explained to the ICT manager—it refers to the art of ensuring the existence and continuity of a nation’s information society, guaranteeing and protecting in cyberspace all of its information assets and critical infrastructure. 

The interaction with the ICT manager was essential for analyzing concepts that influence cyberattacks in telehealth systems, especially in the testimony of their possible consequences associated with the system’s vulnerabilities. The data relevance reinforces the importance of guaranteeing the network’s health since, in the case of loss of confidentiality, it can cause moral damage to all involved, especially to patients [31,32]. Despite many studies identifying threats regarding cybersecurity in distributed systems, there is still a gap in the literature related to the causes that trigger ecosystem cybersecurity occurrences in telehealth systems. 

In addition to the discussion with the information security expert, a total of fifteen variables (concepts) influencing the cyberattacks occurrences in telehealth services were identified, which had support in the literature [33]. These concepts can be considered the weaknesses that affect the operational performance in telehealth systems. Table 1 presents a description of these concepts that the ICT manager has validated, three meetings were held and the time was 1 h. 

The concepts allow complex and critical ecosystem threats to be exploited in a telehealth system. However, the lack or inefficiency of information security planning makes it challenging to identify cybersecurity. This inefficiency also requires tools and methodologies to minimize cybersecurity consequences, which can cause large-scale damage to business sustainability [20]. 

An FCM diagram was built using the ICT manager’s knowledge with the cybersecurity expert’s support through an interview. A cognitive structure with subjective information was generated using the central concepts previously discussed, enabling performance analysis of the telehealth system. This information is associated with the concepts of critical infrastructures—which refers to facilities, services, goods, and systems that will have a severe social and economic impact if their performance is degraded or if they are suspended or destroyed. The visual representation of the expert-based FCM created based on the concepts is shown in Figure 1.

The FCM diagram’s construction aims at verifying the computed values of intensity in the concepts related to cybersecurity in telehealth. The causal relationship between concepts is indicated by an arrow and the positive symbol (+).

The framework of Figure 1 is meant to map the cybersecurity relationships (networks) within the scope of telehealth management by using a Fuzzy Cognitive Map. This process consisted of three phases: 1. Nodes: The key concepts from an Expert Panel; 2. Map: Cause-and-effect relationship in each of the arcs and a graphical representation of the network; and 3. Model: Numerical values and computational simulation. Once the cybersecurity in the telehealth management model is formulated, the subsequent simulation tasks (what-if scenarios) is carried out, with assumptions that modify the input variables (Value Repositories and Constraints), to finally check what impact these changes have on the performance of cybersecurity in the telehealth.

### Outputs of Scenario Analysis

The interpretations of the FCM diagram’s relationships are important for the strategic planning process of the hospital’s ICT department. With these implications, ICT managers can define preference concept actions and develop information security plans capable of minimizing the consequences caused by the vulnerabilities. Each analysis compares the steady-state promoted by the FCM with the scenarios defined by the ICT manager based on the main concepts. Therefore, it is possible to highlight the best and worst scenarios of cyberattacks in the hospital’s telehealth system, considering the concepts of the present study. Table 2 shows the levels of centrality and preferred state for the concepts of cybersecurity in telehealth.

The analysis based on the FCM modeling results allows the ICT manager to build different scenarios of strategic consequences. The construction of the scenarios offers contributions in the simulation of possible implications caused by common factors that affect telehealth systems in a specific way. In addition, these scenarios can support the decision process in the strategic planning of actions to prevent or mitigate vulnerabilities that could compromise the performance of telehealth systems. Planning of mitigation actions, when done without due care can negatively influence the possibility of occurrences of attacks analyzed in Figure 2. The matrix representation of the fuzzy cognitive map (the Wij Weight matrix) obtained after expert interviews and process of modeling change its configuration depending on the experts’ corrections. Based on the current literature, it was found that if a negative value is specified in the initial concept state of the estimation vector, then the modeling results influenced by the factors would be inverted, meaning that hostile factors contribute to cybersecurity.

The main components in telehealth systems, according to ICT expert, judged in the range [−1] to [1], are “Mobile health apps failure” (C3) and “Controls for wireless Communication” (C7) [6]. On the other hand, regarding “Supplier eligibility criteria” (C9) and “Big Data privacy in healthcare” (C11) [46]. Figure 3 illustrates the telehealth scenario analysis.

Scenario I analyzes the impact of the set of the main concept “Mobile health apps failure” (C3) scoring −0.18, “Controls for wireless Communication” (C7) scoring 0.12, “Supplier eligibility criteria” (C9) with scoring −0.07, and “Big Data privacy in healthcare” (C11) scoring −0.51 on the vulnerabilities pointed out in the telehealth system. This scenario highlights the association with the consequence of exploiting vulnerabilities when these factors are identified. These results confirm how changes and wrong configurations can be overflowing the infrastructure of telehealth servers [47,48,49]. 

Further, configurations and composition of the servers responsible for the processing and storage of data and information can increase the probability of attacks that deflect the destination of the data and manipulate the system’s functionalities. Thus, it is necessary to monitor the data origin and destination points, checking what actions are being carried out, as well as to understand the collaboration policies between providers of these ICT services and systems’ users (patients or physicians) so that the university hospital can minimize the damage on the services provided.

In Scenario II, the main components are “Medical System configuration error” (C10) and “IT Investments” (C13). Although each business has its budget destined for investments, procrastinating investment to adequate technology, or using poor quality devices can increase the probability of inefficiency in the answering service and reinforce problems in devices used in telehealth systems. In this context, the effect of cybersecurity is more significant because the malicious action activates defense planning. These situations are generally recorded when the telehealth system comes with records of malware and logical attacks [50]. The analysis related to this Scenario II is represented in Figure 4 and Figure 5.

In Scenario II, as shown in Figure 5, the main concepts are “Sensitive data encryption” (C2 with −0.22) occurrence, “Cybersecurity certification” (C4 with −0.15), “Outsourcing of IT Cloud services” (C5 with −0.14), and “IT Governance” (C6 with −0.06) occurrence. This scenario highlights the concern about controlling the ICT services that are essential for the organization. In medical centers, personal data relating to the patients’ health status should receive greater attention and should be considered requirements for developing specific security policies. The results show that it is possible to view different vulnerability types regarding patient care in the two scenarios. Based on the analysis, it is important to consider that in addition to the value of the information, other criteria must be incorporated in the process of defining the protection requirements of telehealth systems, such as the ability to identify and record system’s threats and vulnerabilities. However, these criteria were not analyzed in the present study. Despite this limitation, it is essential to know in advance the asset’s value to be protected to identify threats and vulnerabilities to return consistent results, which is why cybersecurity planning is needed.

## 4. Discussion

Recent studies argue that the increase in cybersecurity investments has not resulted in more adequate security levels in many areas. This discrepancy can be justified by the lack of consistent information security management [41]. According to Sivaprakash et al. [51], in a comparison made between healthcare and financial organizations, in terms of data management and protection, both types of organizations are concerned and incorporate strategic actions to control and protect data generated in their environments. However, managers do not have adequate training to deal with cyber threats in healthcare organizations [6]. In contrast, financial organizations have been investing in cybersecurity for about twenty years, aligning cybersecurity with the organization’s objectives.

The need for data sharing in heterogeneous public and private healthcare organizations and the lack of continuous and standardized communication in cybersecurity show importance in the responses under the threats and vulnerabilities of the systems, involving medical actors, patients, and ICT analysts [52]. In this context, the ICT professionals have access to data about patients and their clinical status (clinical historic, vital parameters, physical examination data, among other data) that are useful for planning and the decision-making process in telehealth services. However, the provision of healthcare assistance cannot be analyzed as an isolated process but in line with organizational planning as a whole. From this perspective, this study can help the senior manager and the IT manager to understand the vulnerabilities that can affect telehealth systems’ operational performance that contribute as a resource to support cybersecurity planning and ensuring the achievement and enhancement of the efficiency of the information protection in medical centers.

The value of the information is not the only criterion used to define the protection requirements. The measure of the ability to identify threats can be a more consistent indicator of this definition. When the asset’s value is known, the greater the likelihood of efficiency in the process, hence the need for cybersecurity planning. Annual audits, for example, are a way of ensuring minimum compliance with cybersecurity requirements. The determination of an approved regulatory and supervisory body requires organizations to adopt information security procedures and standards to be used as maturity indicators, ensuring an effective cybersecurity policy for telehealth services. The lack of an information security policy is directly reflected in telehealth services’ operational performance.

Our findings show that without imposing any restrictions on cybersecurity, it is possible to allow significant occurrences and negative impacts to reduce telehealth services’ efficiency [53]. The visualization tools allow a better understanding of the causal relationships between the factors and the vulnerabilities considered. FCM is a modeling method for complex systems that use simulations based on the mental map of human reasoning to operate on systems’ representation. Thus, the application of FCM shows the modeling ability to operate ambiguous and vague terms, simulating a sense of words and supporting decision-making and strategic planning of actions related to information security in the health area, a fact reinforced in a previous work of ours (see [54]), which has been expanded by the present article.

### 4.1. University Hospitals and Telehealth Cyber Security Strategies

Regarding the objective unit of the case study, university hospitals, it is noted a strategic decision-making application of actions in an ad hoc stage in relation to cybersecurity risks and necessary measures for prevention and mitigation. This is because, in the university hospital’s perspective where the analysis was applied, planning, information security is considered an essential requirement to be fulfilled within the overall information technology planning. On the other hand, although managers understand the importance of this type of security, it is noteworthy they still do not have the most appropriate tools capable of supporting their decision-making process for related planning, seeking to identify empirically the causal relationships between the various existing elements or concepts, and prioritize them according to their impact on the continuity of telehealth services. At some instance this has been sufficient for mitigating some risks and technological treats.

Resorting to most appropriate tools, however, may offer additional opportunity for managerial continuous improvement. Tools such as FCM, despite popular in many sectors of economic activity and other areas for decision-making, seem to be unknown or underused instruments for cybersecurity managers in Brazil, taking this conclusion specifically within the context of university hospitals. The development of the case study reported here also suggests they can be used relatively easily and efficiently so that these managers can develop plans more in line with the reality they know well, as they develop daily activities on them. Above all, FCM constitute a knowledge management tool capable of externalizing the experiences contained in these managers’ minds, encoding this experience in an intelligible and accessible way for use in cybersecurity and information security planning.

### 4.2. Comparison with Other Methods/Approaches Found in Literature

In Table 3, a synthesis of the works containing similar methods used in the literature to support the development of this article will be presented. It contains the objectives and main similarities and differences, as well as a synthesis of this work, for comparative purposes.

## 5. Conceptual and Practical Implications

Our results highlight cybersecurity issues in telehealth services that deserve special attention, whether from a conceptual or practical point of view since sensitive data circulates through any type of information system. In this sense, exploring the system’s possible vulnerabilities is fundamental to adopting preventive or corrective measures [55].

This issue is even more delicate in telehealth services and systems since certain information may be under medical confidentiality and can compromise patients’ physical and psychological integrity should they be improperly exposed [56]. The conceptual point of view about using FCM in telehealth systems is linked to how this tool can influence the planning and adoption of security measures in these systems. Here we can establish the following question: how should these systems be thought of, from their planning through their implementation, finally reaching their full functioning, to ensure that this sensitive information is protected efficiently and effectively?

Our model demonstrates that several concepts related to threats in systems and types of cyberattacks, always considering the participation of experts, whose understanding of the relationships between these concepts is represented through the graph resulting from the application of the FCM. These relationships are still supported by obtaining a measure of strength extracted from a fuzzy context that represents vagueness in the definitions made by these experts when eliciting his knowledge.

Here it can be connected with knowledge engineering, which states that eliciting or extracting expert (tacit) knowledge is a bottleneck and a critical issue in systems development [57]. In this analogy, the FCM acts as a formal means for this knowledge to be acquired and recorded, allowing the engineers and systems analysts involved with telehealth systems projects to correct existing security breaches and design plans for action contingency of possible cyberattacks.

From the perspective of telehealth systems actors, whether health professionals or patients, concepts such as confidentiality, consistency, and availability of information, together with the use of these systems only by authorized personnel and the presence of functions to reduce errors [58], deserve mention in this discussion, to add or enforce security requirements. In addition to the professionals responsible for designing, implementing, and managing information systems and ensuring information security, users also deserve to be heard since they are the final subjects to whom the system was designed [10].

Therefore, the applied methodology can be extended to obtain new security perceptions about the telehealth system, reinforcing those already elicited from experts. These two perspectives, in fact, require feedback: (i) on the knowledge of experts providing technical elements for the design and implementation of systems and information security measures; and (ii) on the opinion of the end-users, being evaluated based on these technical elements, to reinforce them or identify new requirements.

Based on our empirical results, referring to vulnerabilities, forms of cyber-attacks, and user concerns, can be analyzed through the FCM. The results obtained should be discussed by the security project team in a post-conceptualization stage. While other works have their approach focused on more technical elements related to the security guarantee in telehealth systems, the value of the methodology used in this work is at a more managerial and strategic level, ensuring the visualization of the related concepts for making decisions about themselves.

This part of the discussion aims to determine what should be implemented as a priority since the conceptual elements detected through the methodology are likely to be in large numbers. Trade-offs will emerge in this type of valuation, such as less time spent on systems valuation instead of information and more time spent assigning values to the assets involved [59].

The following question arises: what is the most appropriate way to evaluate these concepts and to choose what will be implemented as a priority? Each team must carry out the evaluations according to what is defined by the organization, and the users’ opinions deserve attention, complementing the information security requirements. Nevertheless, it is essential to note that the information collected mainly from the users can provide valuable feedback to the project development team. FCMs have the advantage of showing defuzzified numerical values referring to the relationships between the evaluated elements [60,61]. These indicators can be combined with other more common elements in evaluating alternatives to be implemented, such as the cost and time involved. Moreover, FCMs make it possible for decisions to be made by analyzing varied scenarios built based on the subjective opinions of the people involved [31], ensuring the inclusion of elements described in a technical and non-technical way, the latter related to the perspectives of users.

Furthermore, practical implications at a higher level, leaving aside the view on more technical elements, the use of FCM favors the creation of information security and cybersecurity policies. Analyzing the existing relationships between guidelines, requirements, and rules—elements that constitute these policies and lead to information security compliance [55]—is a process potentially facilitated by using the explored methodology. On the other hand, the definition of these policies implies the determination of a pattern of user behavior towards security in telehealth systems, since the behavioral factor alone has a more considerable influence than technical security elements in related systems and services [62] since the focus of the analysis now becomes the users’ conduct as a “breach breaker” of security in the system.

In summary, the practical implications of the use of FCMs fall on implementing the telehealth system, providing security requirements to be implemented, whether defined by the experts’ perspective or considering users’ opinions. Also, the conduct of users of the system must be in line with security policies, which are also definitions that can be carried out with the support of the methodology.

## 6. Conclusions

In general, telehealth, precisely its technological, economic, and environmental characteristics, substantially contribute to society and is expected to provide health services to thousands of people limited by geographical constraints. Given this context, telehealth can benefit from the scenario-planning approach because it plays an essential role in future development related to planning policies against cyberattacks.

This paper presented an application of FCM that analyzes information security factors related to telehealth. The FCM model allowed the causal inference of direct chaining and numerical data-based updates and cybersecurity experts’ opinions. Preliminary results are encouraging concerning the FCM approach’s possibilities to decision-makers/ICT managers, enabling a good insight into the impact of cyberattacks on telehealth and ensuring a more focused view of the necessary protective actions. These results show the possibility of obtaining scenario planning in cybersecurity, highlighting the most critical telehealth factors. The analytical process should be carried out annually or semiannually to analyze the impact of improvements in information security, with possible improvements addressing identified critical points.

Although our focus is on the main concepts of aligning cybersecurity in telehealth, it should be noted that the construction of FCM allowed the identification of new concepts. In particular, the problem of image privacy of medical exam results can affect patients’ integrity. Moreover, new concepts were included in the FCM, as they are rarely considered in security practice, which allowed it to be formalized in a way that contributed to reducing the variables omitted in the decision on cybersecurity.

The tools proposed by previous FCM literature were suitable for the cybersecurity scenario due to the ability to capture the ICT experts’ knowledge by modeling dynamic simulation systems and improving support against cyberattacks. COVID-19 has dramatically impacted telehealth functionality and required adaptation in coping with circumstances that continued to change relative to safety measures, limiting customer interactions and reducing employee availability. The COVID-19 pandemic has generated remarkable and unique societal and economic events leveraged by cyber-criminals. Our analysis of telehealth has shown the causes of cybersecurity in telehealth services.

With many new perspectives brought by the current pandemic, we believe this new paradigm for cybersecurity in telehealth also came to stay in the post-pandemic (hopefully) new future. FCMs can be adjusted according to iterative scenarios to support accurate decision-making representing subjectivity in the business model of healthcare units. In addition, it can increase the transparency of analyses, including information hidden to IT managers. The post-pandemic is an important consideration to accommodate many legal aspects generated during the pandemic, specially related to the computerization of various services or intensification of current computerized services, as it is the case with telehealth. Therefore, this kind of application is essential for helping hospital managers concerned with the maintenance of telemedicine services during the planning phases, which are not limited to the pandemic context.

It is worth noting that telemedicine has become an efficient and effective way to develop the necessary care in a critical period such as the COVID-19 pandemic, avoiding hospital overload with high demands of patients seeking care, and avoiding contamination by the disease amidst clusters of people. Our perception leads us to believe that cybersecurity measures in telehealth systems have entered as mandatory components in ICT planning for hospital institutions, ensuring the security of patient information and ensuring that services continue to run without interruptions and external interference, such as hacker attacks. The FCMs are a helpful instrument for university hospital managers concerned with the maintenance of their telemedicine services, and regardless of the pandemic context, they deserve to be applied in the associated planning phase.

Therefore, the added value of using FCMs in cybersecurity in telemedicine is none other than supporting the planning of strategies to combat security breaches, always preventing sensitive and sensitive patient information from being accessed or intercepted by inappropriate persons. In the planning practice, it is a new tool for managers to use, in the planning practice, helping in their decisions about actions to avoid or correct security problems.

Future work should aggregate other methods to assist ICT managers in deciding upon actions such as using fuzzy sets theory to translate the judgments of health units’ managers into crisp values for an accurate support that can minimize cybersecurity problems in telehealth [63], and combining multicriteria methods with other operational methodologies for conflict resolution, resource management and risk assessment in telemedicine [53,55,64]. More specifically, the research leading to this article, for the time being, has implications for the construction and improvement of a framework aimed at identifying risks associated with cybersecurity in telemedicine, carrying out tests for its validation in Brazilian university hospitals.

Concerning the continuation of this research, it is possible to define the need to assess how university hospitals, in a study of multiple cases, are prepared to deal with cybersecurity threats, clarifying what the main strategies adopted are, in addition to how the planning process is developed for these strategies, gathering data with a set of these hospitals. Another indication is the development of a meta-analysis study comparing quantitative results of other works containing methods applied with the same purpose as the one applied in this study, helping mainly to determine which methods are most suitable to support the planning process in cybersecurity in telehealth.

For these two last indications of further research, as we did not aimed at evaluating a general context for cybersecurity, and evaluating the performance of many different healthcare institutions to know how well they are in preparing to face telehealth cybersecurity threats, they are beyond of the scope of our current application. Therefore, these limitations can be addressed in future extensions of the current analysis.

## Figures and Tables

**Figure 1 healthcare-09-01504-f001:**
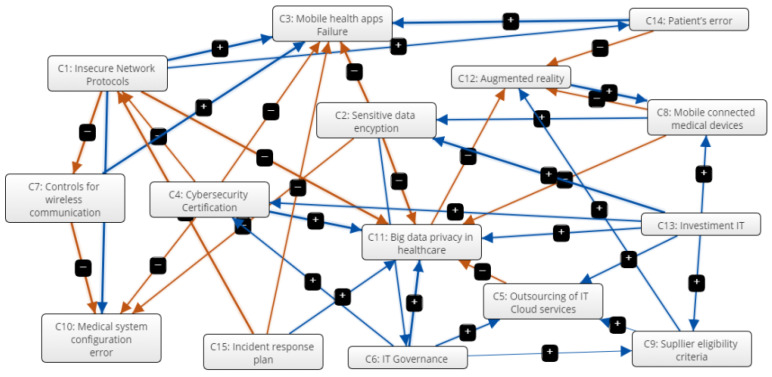
Model FCM cybersecurity in the telehealth university hospital.

**Figure 2 healthcare-09-01504-f002:**
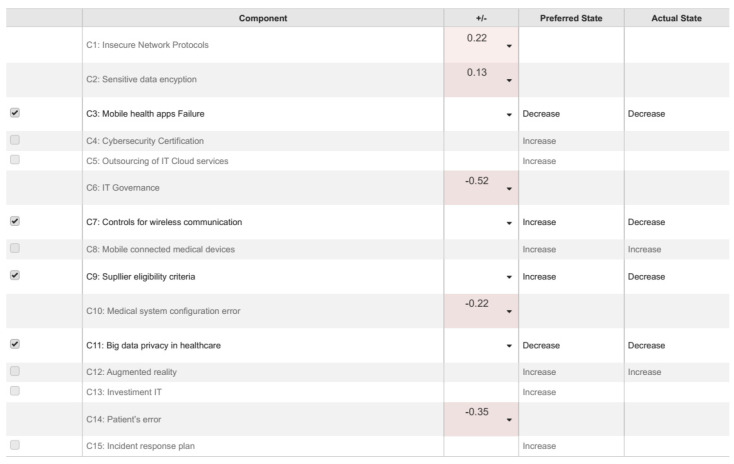
Final equilibrium states by the value of nodal element C1 (insecure network protocols), C2 (Sensitive data encryption), C6 (IT Governance), C10 (Medical system configuration error), and C14 (Patient’s error).

**Figure 3 healthcare-09-01504-f003:**
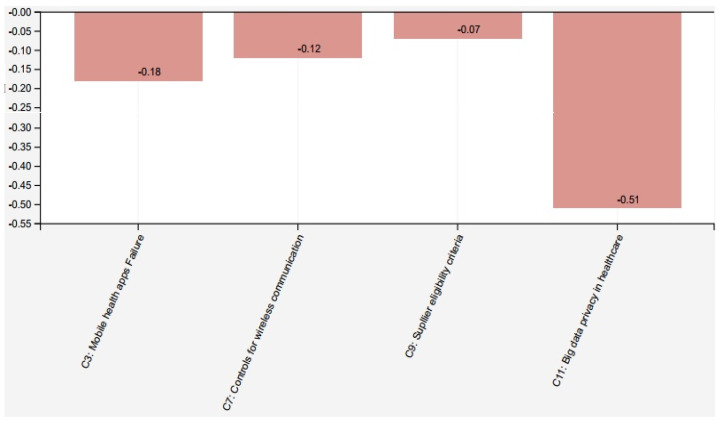
Scenario I: analysis cybersecurity in telehealth.

**Figure 4 healthcare-09-01504-f004:**
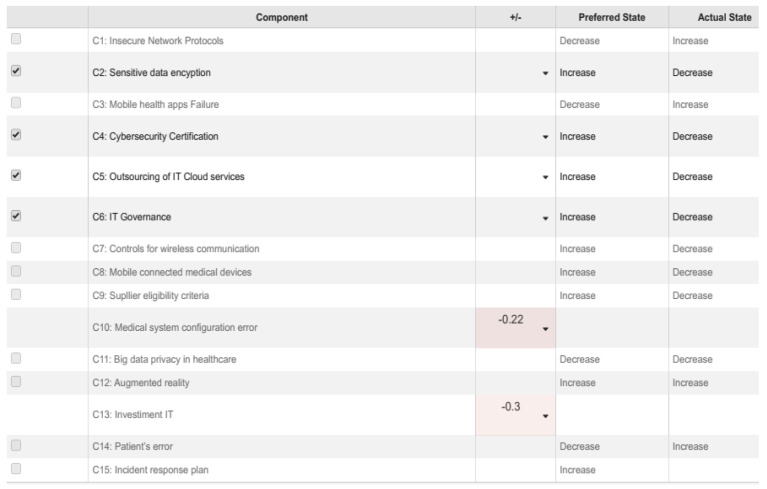
Final equilibrium states by the value of nodal element C10 (Medical system configuration error) and C13 (IT Investment).

**Figure 5 healthcare-09-01504-f005:**
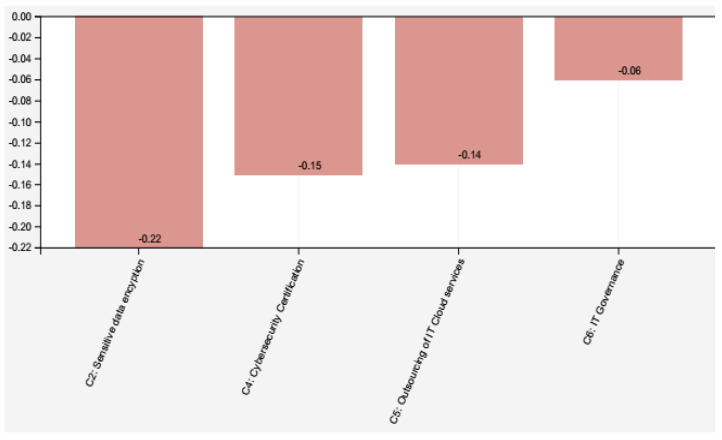
Scenario II: analysis cybersecurity in telehealth.

**Table 1 healthcare-09-01504-t001:** Description of variables involved in the study in telehealth services.

Main Concepts	Description	Fuzzy Interpretation	References
C1: Insecure network protocols	Due to insecure network protocols, (HTTP), attackers can enter the organization’s network	−1: Low incompatibility network protocol 0: Average incompatibility network protocol 1: High incompatibility network protocol	[34]
C2: Sensitive data encryption	Involve custom code development that brings encryption into the individual application data fields	−1: Low Information Security maintenance 0: Average Information Security maintenance 1: High Information Security maintenance	[35]
C3: Mobile health apps failure	Operational failures occur in telehealth due to users not being prepared to adopt information security protocols.	−1: Low Operational failures occur in telehealth 0: Average Operational failures occur in telehealth 1: High Operational failures occur in telehealth	[36]
C4: Cybersecurity certification	Provides a rationale for why the auditable events are deemed to be adequate to support the after-fact investigations of security incidents into operational telehealth server	−1: Absolute abandonment of auditable events. 0: Average attention to auditable events. 1: Priority attention to auditable events	[37]
C5: Outsourcing of IT cloud services	Provides help desks, tech support, and provider to protect the confidentiality of the outsourced information.	−1: No supporting communication security. 0: A few supporting communication security. 1: Priority attention to communication security	[38]
C6: IT governance	Provides security strategies aligned with and supporting the business objectives	−1: Absolute abandonment of IT Governance. 0: Average attention to IT Governance. 1: Priority attention to IT Governance	[39]
C7: Controls for wireless communication	Establishment of policies and procedures for the effective implementation of selected security and control enhancements into telehealth.	−1: Absolute abandonment of policy access. 0: Average attention to policy access. 1: Priority attention to policy access	[40]
C8: Mobile connected medical devices	Lack of updates or lack of patching, a common threat that can have a significant impact on the healthcare organization	−1: Low Information Security maintenance 0: Average Information Security maintenance 1: High Information Security maintenance	[5]
C9: Supplier eligibility criteria	Establish security baseline requirements and translate them into eligibility criteria when selecting suppliers	−1: No supporting supplier eligibility 0: A few supporting Supplier eligibility 1: Plenty of supporting supplier eligibility	[41]
C10: Medical system configuration error	Medical platforms are software that needs to be installed on a practice or health system’s local server	−1: No supporting medical systems. 0: A few supportive medical systems. 1: Priority attention of medical systems.	[42]
C11: Big data privacy in healthcare	Big data has considerable potential to improve patient outcomes and predict outbreaks of epidemics	−1: Low Information Security maintenance 0: Average Information Security maintenance 1: High Information Security maintenance	[43]
C12: Augmented reality	Provide remote clinicians, such as surgeons, to guide physicians, paramedics, and other staff to perform emergency procedures in telehealth	−1: No supporting augmented reality 0: A few supporting augmented reality 1: Plenty of supporting augmented reality	[44]
C13: IT Investment	Provides IT investments during the pandemic, accelerating the use of telemedicine services	−1: No supporting IT Investment 0: A few supporting IT Investment 1: Plenty of supporting IT Investment	[35]
C14: Patient’s errors	Providers should educate patients about cybersecurity and the steps they should take to improve the overall safety of their interactions online	−1: No supporting education. 0: A few supporting education. 1: Plenty of supporting education	[45]
C15: Incident response plan	Systems and devices eventually fail due to inaccurate coding, improper handling, or just tear and wear	−1: No supporting incident plan. 0: A few supporting incident plan. 1: Plenty of supporting incident plans	[6]

**Table 2 healthcare-09-01504-t002:** Degree of the centrality of IT manager preference concepts.

Main Concepts Cybersecurity in Telehealth	Indegree	Outdegree	Centrality	Preferred State
C1: Insecure network protocols	1.01	2.69	3.71	Decrease
C2: Sensitive data encryption	0.95	1.88	2.83	Increase
C3: Mobile health apps failure	3.26	0.00	3.25	Decrease
C4: Cybersecurity certification	0.65	1.67	2.32	Increase
C5:Outsourcing of IT cloud services	0.88	0.33	1.22	Increase
C6: IT governance	0.27	1.34	1.61	Increase
C7: Controls for wireless communication	0.56	1.05	1.61	Increase
C8: Mobile connected medical devices	0.91	0.97	1.88	Increase
C9: Supplier eligibility criteria	0.41	0.35	0.77	Increase
C10: Medical system configuration error	1.68	0.00	1.68	Decrease
C11: Big Data privacy in healthcare	3.82	0.34	4.17	Decrease
C12: Augmented reality	1.10	0.52	1.62	Increase
C13: Investments IT	0.00	2.39	2.39	Increase
C14: Patient’s error	0.32	0.89	1.13	Decrease
C15: Incident response plan	0.00	1.48	1.48	Increase

**Table 3 healthcare-09-01504-t003:** Literature comparison.

Reference	Objective	Main Similarities	Main Differences
[10]	Develop and validate a telehealth privacy and security self-assessment questionnaire to be applied with providers.	It applies expert assessment that can be used to identify vulnerabilities in telehealth systems.	It does not establish causal relationships among the identified elements. The applied procedure is based in the application of questionnaires and psychometric analysis.
[12]	Present a big data risk model using Failure Mode and Effects Analysis (FMEA) and Grey Theory.	It provides a structured approach to assess risk factors, facilitating the assessment and providing a vision of risks relations.	The work uses Different methods (Failure Mode and Effects Analysis and Grey Theory).
[13]	Propose a risk model for information security that identify and evaluate the events’ sequence in scenarios related to the abuses of information technology systems.	The model allows ranking the risks based on their criticality, supporting the definitions of preventive or corrective actions. Use of Fuzzy Theory elements.	It does not establish causal relationships among the identified elements. Use of Event Tree Analysis.
[14]	Propose an approach to information security risk management based on Failure Mode and Effects Analysis (FMEA) and Fuzzy Theory.	The approach applies identification of risk elements/concepts, prioritizing risk dimensions according to the risk’s criticality, to support defining preventive or corrective actions. Use of Fuzzy Theory elements.	It does not establish causal relationships among the identified elements. Use of Failure Mode and Effects Analysis.
[15]	Propose a model to evaluate cybersecurity risk using Fault Tree Analysis, Decision Theory and Fuzzy Theory.	The model analyses risk scenarios also using elements from Fuzzy Theory, supporting the identification of vulnerabilities in cybersecurity linking them with potential consequences.	Use of Fault Tree Analysis, with elements from decision theory.
[23]	Propose a framework for cybersecurity risk management in telemedicine.	Identification of causes, consequences, and preventive measures for security threats, using scenario analysis.	Different methods (fault tree analysis and event tree analysis).
[29]	Propose a quantitative assessment framework to evaluate nuclear power plant risks related to cyber-attacks.	Assessment of cybersecurity risk elements, using scenarios, and providing risk information to develop preventive or corrective strategies.	Use of difficulty and consequences of cyber-attacks in the assessment, use of Bayesian belief networks and probabilistic safety assessment methods.
	Objective	Main characteristics	Main characteristics compared to other models/approaches
This work	Propose an analytical approach using Fuzzy Cognitive Maps (FCM) representing experts’ opinions about causal relationships of concepts related to cybersecurity in telehealth systems, providing support for strategic planning and decision-making	Use of expert knowledge creating a graphical representation about expert reasoning about cybersecurity threats, aiding to prioritize them according to scenarios. Support to cybersecurity strategies development by understanding the causal relationships between the concepts.	The approach applied in this study do not consider the probabilistic component involved in risk analysis, in its mathematical formulation to generate de graphs from FCM. Most of the methods or approaches previously presented deal with probabilistic data about the security threats.

## Data Availability

The data presented in this study are available on request from the corresponding author.

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
