# Peer review of "Fuzzy Cognitive Scenario Mapping for Causes of Cybersecurity in Telehealth Services"

_healthcare, 2021, doi:10.3390/healthcare9111504_

Round 1

Reviewer 1 Report

This paper presented an application of FCM that analyzes information security factors related to telehealth. The study is significant.

There are some problems in paper content organization and method description:

  1. In ABSTRACT, Line 1-7 is repeated, and maybe some sentences are missed.
  2. The second section is only 2.1, in addition, the content is not completely consistent with the title, and there is no description of the characteristics of the source data, nor of the mapping and fusion method.
  3. There are two titles named as ‘4.1. Conceptual Implications’.
  4. It is better to give some discussion on relation works of Cognitive Scenario Mapping in Introduction.

Author Response

This paper presented an application of FCM that analyzes information security factors related to telehealth. The study is significant.

There are some problems in paper content organization and method description:

  1. In ABSTRACT, Line 1-7 is repeated, and maybe some sentences are missed.

Authors’ Answer: The issue was corrected, and the entire Abstract was reviewed in the process.

  1. The second section is only 2.1, in addition, the content is not completely consistent with the title, and there is no description of the characteristics of the source data, nor of the mapping and fusion method.

Authors’ Answer: Thank you for the comment. We kept only a Materials and Methods section, with no subsections. We believe that the text sequence within this section already fully describes the adopted methodological procedure.

  1. There are two titles named as ‘4.1. Conceptual Implications’.

Authors’ Answer: We have transformed the two subsections within Discussion Section 4 into a Section 5 called Conceptual and practical implications.

  1. It is better to give some discussion on relation works of Cognitive Scenario Mapping in Introduction.

Authors’ Answer: The importance of considering vulnerabilities is often associated with the risk of losses, corruptions, inappropriate changes and theft of data, information and documents that affect the integrity of medical diagnoses delivered to the patient, which can cause serious damage to the health of the individual (Sun et al. 2020). In general, these situations allow threats to be exploited and are often caused by cyberattacks from malicious systems or people (Silva et al. 2014). (Zain and Clarke 2005) identified four main situations verified in cyberattacks which can occur in telehealth services, such as (i) when the data is destroyed or becomes unavailable, (ii) when an unauthorized system or person access the database, (iii) when an unauthorized system or person gets access to the service and makes improper changes, and (iv) when an unauthorized system or person insert counterfeit objects into the database. According to (Zain and Clarke 2005), these situations are possible failures or threats in data transmission process, which can be accidental or purposeful.

In telehealth services, the main challenge of the physicians is protecting the privacy of data. However, most of these professionals do not receive adequate training, and they are subject to situations that may compromise the performance of healthcare. This context requires preventive actions and security tools due to the sensitive data in healthcare systems such as digital signatures, professional’ credentials, financial data, patient diagnostic images, among others (Faragallah et al. 2020). It is worth mentioning that this concern becomes even more complex when considering cyberattacks, especially due to the different interactions that occur on the Internet (Lim, 2008). Furthermore, failure to comply with legal regulations may result in financial, criminal penalties (Andriole 2014; Nagasubramanian et al. 2018). For this, the IT professionals must make strategic decisions to define security policies and ensuring authenticity, integrity, and confidentiality of the database, besides ensuring business sustainability.

Most of these strategic decisions are involved in business sustainability process (Barney and Hesterly 2015), which can define action plans to ensure the telehealth services operation. The ICT management process assists in directing how medical centers can use IT to manage technologic solutions. For this, it is opportune to present methodologies to support organizational diagnoses to identify these possible causes of threats in telehealth systems. One of these methodologies is Fuzzy Cognitive Maps (FCMs) (Kosko 1986) which represents scientific knowledge and strategic decision making in systems using elements of a mental map, based on fuzzy logic computation.

Reviewer 2 Report

The research topic is relevant and attractive to society. Below are the comments to the authors.

There are duplicated ideas in the abstract, it is recommended to adjust the wording.

Typos were identified on the lines,  

  • wi and wj
  • only be accessed on the network, thus
  • C2: Sensitive data encyption

In a technical o scientific context there are some phrases that could be imprecise, it is recommended to review them.

  • This process was essential to specify the vulnerabilities that the IT manager could identify.
  • When information security planning is inefficient, there is a high probability of insertion of vulnerabilities into the telehealth system.
  • The concepts can directly affect a vulnerability in the system
  • When security policies cannot predict threats and cyberattacks.
  • The present work identified cyber vulnerabilities based on potential weakness factors to telehealth system´s defenses, demonstrating how cyberattacks in adverse moments tends to compromise a system´s stability.

Scientific comments about the paper  

The research motivation is based on the question, how are university hospitals preparing to face telehealth cybersecurity strategies? I did not find the answer section to the research question.

The model presented in figure 1 is fundamental to research´s contribution, however, the article does not describe the process for its definition. Similarly occurs with the results shown in tables 2 and 3.

The review of the state of the art was not identified, in this sense, some questions arise:

what are the advantages and disadvantages of the research contribution compared to previous proposals on risk analysis, vulnerability assessment and critical assets identification?

Could the results shown in tables 2, 3 and in figure 2 be compared with the state of the art?

Derived from the review of the article I have other types of questions

Table 1 presents concepts related to cybersecurity and others that are not necessarily, what were the criteria to validate the concepts? Why are only wireless communication controls mentioned? a security architecture is made up of a much broader universe.

Do the concepts presented represent a completed security architecture?

The assets presented are the complete universe for a university hospital?

Are there differences between a university hospital and other types of hospital?

The attack surface is a fundamental element in security, failure are elements of a different area of knowledge, what is the relationship between the concepts in the context of the work?

How are vulnerabilities classified in research?

In figure 3 shown the scenario I, did you consider all kind of vulnerabilities?

Why does feedback from non-governance or cybersecurity experts complement security requirements?

Author Response

The research topic is relevant and attractive to society. Below are the comments to the authors.

There are duplicated ideas in the abstract, it is recommended to adjust the wording.

Typos were identified on the lines,  

  • wi and wj
  • only be accessed on the network, thus
  • C2: Sensitive data encyption

Authors’ Answer: Thank you for the comment. We have corrected the errors indicated.

In a technical o scientific context there are some phrases that could be imprecise, it is recommended to review them.

  • This process was essential to specify the vulnerabilities that the IT manager could identify.
  • When information security planning is inefficient, there is a high probability of insertion of vulnerabilities into the telehealth system.
  • The concepts can directly affect a vulnerability in the system
  • When security policies cannot predict threats and cyberattacks.
  • The present work identified cyber vulnerabilities based on potential weakness factors to telehealth system´s defenses, demonstrating how cyberattacks in adverse moments tends to compromise a system´s stability.

Authors’ Answer: We removed all the unclear sentences, as we observed that they did not have any contribution to enrich the explanations in the paragraphs where they were.

Scientific comments about the paper

The research motivation is based on the question, how are university hospitals preparing to face telehealth cybersecurity strategies? I did not find the answer section to the research question.

Authors’ Answer: Thank you for this comment. A stable telehealth sector should ensures information security for the patient. Te main requirement of any hospital is security. The analysis based on the FCM modeling results allows the ICT manager to build different scenarios of strategic consequences. The scores from the scenario analysis of section “3. Results” is associated with this motivation which the reviewer raise attention. We hope this is sufficient, but we remain available for additional discussions.

The model presented in figure 1 is fundamental to research´s contribution, however, the article does not describe the process for its definition. Similarly occurs with the results shown in tables 2 and 3.

Authors’ Answer:  Thank you for this valuable comment. The framework of Figure 1 is meant to map the cybersecurity relationships (networks) within the scope of telehealth management by using a Fuzzy Cognitive Map. This process consisted of three phases: 1. Nodes: The key concepts from an Expert Panel. 2. Map: Cause-and-effect relationship in each of the arcs and a graphical representation of the network and 3. Model: Numerical values and computational simulation. Once the cybersecurity in the telehealth management model is formulated, the subsequent simulation tasks (what-if scenarios) is carried out, with assumptions that modify the input variables (Value Repositories and Constraints), to finally check what impact these changes have on the performance of cybersecurity in the telehealth. We have included this in the seventh paragraph of section 3, and rewritten the manuscript where appropriate for a better description of results.

The review of the state of the art was not identified, in this sense, some questions arise:

what are the advantages and disadvantages of the research contribution compared to previous proposals on risk analysis, vulnerability assessment and critical assets identification?

Authors’ Answer: Authors added new discussion: The importance of considering vulnerabilities is often associated with the risk of losses, corruptions, inappropriate changes and theft of data, information and documents that affect the integrity of medical diagnoses delivered to the patient, which can cause serious damage to the health of the individual (Sun et al. 2020). In general, these situations allow threats to be exploited and are often caused by cyberattacks from malicious systems or people (Silva et al. 2014). (Zain and Clarke 2005) identified four main situations verified in cyberattacks which can occur in telehealth services, such as (i) when the data is destroyed or becomes unavailable, (ii) when an unauthorized system or person access the database, (iii) when an unauthorized system or person gets access to the service and makes improper changes, and (iv) when an unauthorized system or person insert counterfeit objects into the database. According to (Zain and Clarke 2005), these situations are possible failures or threats in data transmission process, which can be accidental or purposeful.

In telehealth services, the main challenge of the physicians is protecting the privacy of data. However, most of these professionals do not receive adequate training, and they are subject to situations that may compromise the performance of healthcare. This context requires preventive actions and security tools due to the sensitive data in healthcare systems such as digital signatures, professional’ credentials, financial data, patient diagnostic images, among others (Faragallah et al. 2020). It is worth mentioning that this concern becomes even more complex when considering cyberattacks, especially due to the different interactions that occur on the Internet (Lim, 2008). Furthermore, failure to comply with legal regulations may result in financial, criminal penalties (Andriole 2014; Nagasubramanian et al. 2018). For this, the IT professionals must make strategic decisions to define security policies and ensuring authenticity, integrity, and confidentiality of the database, besides ensuring business sustainability.

Most of these strategic decisions are involved in business sustainability process (Barney and Hesterly 2015), which can define action plans to ensure the telehealth services operation. The ICT management process assists in directing how medical centers can use IT to manage technologic solutions. For this, it is opportune to present methodologies to support organizational diagnoses to identify these possible causes of threats in telehealth systems. One of these methodologies is Fuzzy Cognitive Maps (FCMs) (Kosko 1986) which represents scientific knowledge and strategic decision making in systems using elements of a mental map, based on fuzzy logic computation.

Could the results shown in tables 2, 3 and in figure 2 be compared with the state of the art?

Authors’ Answer: Thank you for the comment. We have added the following (see section 3, pg 7): “The analysis based on the FCM modeling results allows the ICT manager to build different scenarios of strategic consequences. The construction of the scenarios offers contributions in the simulation of possible implications caused by common factors that affect telehealth systems in a specific way. In addition, these scenarios can support the decision process in the strategic planning of actions to prevent or mitigate vulnerabilities that could compromise the performance of telehealth systems. Planning of mitigation actions, when done without due care; can negatively influence the possibility of occurrences of attacks analyzed in Figure 2. The matrix representation of the fuzzy cognitive map (the Wij Weight matrix) obtained after expert interviews and process of modeling change its configuration depending on the experts’ corrections. Based on the current literature, it was found that if a negative value is specified in the initial concept state of the estimation vector, then the modeling results influenced by the factors would be inverted, meaning that hostile factors contribute to cybersecurity.”

Derived from the review of the article I have other types of questions

Table 1 presents concepts related to cybersecurity and others that are not necessarily, what were the criteria to validate the concepts? Why are only wireless communication controls mentioned? a security architecture is made up of a much broader universe.

Authors’ Answer: Thank you for the comment. Building a comprehensive FCM model was required the inclusion of main factors of cybersecurity telehealth occurrence. For developing an FCM that models the causality relations of accident occurrence, the division of FCM concepts into cybersecurity and telehealth is necessary for obtaining practical analysis ability and better visualization for the map. Narrowing the focused concepts to the most central and uncertain items places value on the analysis when there are too many parameters. In recognizing the key concepts of cognitive maps, the professional’ points of view are incorporated into the characteristics of the methods.

Do the concepts presented represent a completed security architecture?

Authors’ Answer:  Thank you very much for the question. Yes. FCM development begins with defining the objective and the scope of the study. The objective and the scope both guide stakeholder identification and guide the questions posed to stakeholders to elicit knowledge on the system. The scope refers to the study area the FCM aims to describe. Delineating the scope of the security architecture is important as discussions at different levels yield different results. For instance, describing a system at the cybersecurity level will yield a different set of concepts from describing a system on a larger level such as the national level. Then, integrating multiple perspectives in understanding a complex system is highly dependent on the participating stakeholder, which makes stakeholder selection very important. Stakeholder involvement alone is not enough to satisfy that a PM process took place. At the start of the interview, the objective and the scope of the study were explained to the stakeholder. Stakeholders were briefed to consider as wide a range as possible of concepts/factors/drivers including social, economic and environmental factors influencing. we collated a list of concepts mentioned in all stakeholder interviews. We further analyzed these concepts by clustering similar concepts/terms together. To support this aggregation, we conducted a content analysis of scientific publications in the field of rice that refer to the hospital. Thereafter we analyzed the statements made by stakeholder one after the other to establish connections. For this process, the content analysis of literature provided the commonly used terms in literature. Stakeholders participated in a 2nd episode by completing an online form. Stakeholders are presented with connections as pairwise relationships. FCMs can be considered representations of pairwise associations using qualitative terms which we convert to quantitatively assigned weighted edges between − 1 and 1.

The assets presented are the complete universe for a university hospital? Are there differences between a university hospital and other types of hospital?

Authors’ Answer: In Brazil, university hospitals work with the training of human resources and development of technology for the area and health. In essence, they provide the same health service as any other healthcare units. In addition, the effective provision of services to the population university hospitals allows for the constant improvement of care and the development of technical protocols for the various pathologies. This ensures better standards of efficiency, available to the Unified Health System (SUS) network.

The attack surface is a fundamental element in security, failure are elements of a different area of knowledge, what is the relationship between the concepts in the context of the work?

Authors’ Answer: Thanks for your consideration. A description of the initial situation, a set of concepts (target and control factors), edges (control vectors) and their levels of influence, formed by the authors and experts, depending on the scenarios that determine the effective control factors. For clarity, a graphical representation of the resulting FCM is formed. According to the process of developing fuzzy cognitive maps, the number and type of concepts were defined by the expert.

How are vulnerabilities classified in research?

Authors’ Answer: Thank you very much for the comment. We appreciate your valuable and considerable feedback on the manuscript .Vulnerabilities are flaws that can be exploited by a malicious entity to gain greater access or privileges than it is authorized to have on a computer system. Not all vulnerabilities have related patches; thus, system administrators must not only be aware of applicable vulnerabilities and available patches, but also other methods of remediation (e.g., device or network configuration changes, employee training) that limit the exposure of systems to vulnerabilities.

In figure 3 shown the scenario I, did you consider all kind of vulnerabilities?

Authors’ Answer: In this figure, the experts considered the impact of the following vulnerabilities on the telehealth structure: “Mobile health apps failure” (C3), “Controls for wireless Communication” (C7), “Supplier eligibility criteria” (C9), and “Big Data privacy in healthcare. From this analysis, new factors can be identified and studied in the context of telehealth.

Why does feedback from non-governance or cybersecurity experts complement security requirements?

Authors Answer: Thank you very much for the comment. The lack of governance has serious repercussions for both privacy by design as well as the democratic governance of technologies for public use, matters pertinent to reflect on given the increasing number of applications for telehealth management using digital technologies.

Round 2

Reviewer 1 Report

The work of this paper is  innovative.

The paper has been sufficiently improved, I think it can be accepted for publishing.

Author Response

Thank you!